# Development of novel g-SSR markers in guava (*Psidium guajava* L.) cv. Allahabad Safeda and their application in genetic diversity, population structure and cross species transferability studies

Chavlesh Kumar[1], Ramesh Kumar[2], Sanjay Kumar Singh[1], Amit Kumar Goswami[1], A. Nagaraja[1], Ritu Paliwal[2], Rakesh Singh[2]*

**1** Division of Fruits and Horticultural Technology, ICAR-Indian Agricultural Research Institute, New Delhi, India, **2** Division of Genomic Resources, ICAR-National Bureau of Plant Genetic Resources, New Delhi, India

* singhnbpgr@yahoo.com

**Data Availability Statement:** All relevant data are within the paper.

## Abstract

Dearth of genomic resources particularly, microsatellite markers in nutritionally and commercially important fruit crop, guava necessitate the development of the novel genomic SSR markers through the library enrichment techniques. Three types of 3'-biotinylated oligonucleotide probes [$(CT)_{14}$, $(GT)_{12}$, and $(AAC)_8$] were used to develop microsatellite enriched libraries. A total of 153 transformed colonies were screened of which 111 positive colonies were subjected for Sanger sequencing. The clones having more than five motif repeats were selected for primer designing and a total of 38 novel genomic simple sequence repeats could be identified. The g-SSRs had the motif groups ranging from monomer to pentamer out of which dimer group occurred the most (89.47%). Out of 38 g-SSRs markers developed, 26 were found polymorphic, which showed substantial genetic diversity among the guava genotypes including wild species. The average number of alleles per locus, major allele frequency, gene diversity, expected heterozygosity and polymorphic information content of 26 SSRs were 3.46, 0.56, 0.53, 0.29 and 0.46, respectively. The rate of cross-species transferability of the developed g-SSR loci varied from 38.46 to 80.77% among the studied wild *Psidium* species. Generation of N-J tree based on 26 SSRs grouped the 40 guava genotypes into six clades with two out-groups, the wild guava species showed genetic distinctness from cultivated genotypes. Furthermore, population structure analysis grouped the guava genotypes into three genetic groups, which were partly supported by PCoA and N-J tree. Further, AMOVA and PCoA deciphered high genetic diversity among the present set of guava genotypes including wild species. Thus, the developed novel g-SSRs were found efficient and informative for diversity and population structure analyses of the guava genotypes. These developed novel g-SSR loci would add to the new genomic resource in guava, which may be utilized in genomic-assisted guava breeding.

**Funding:** The authors did not receive specific funding for this work.

**Competing interests:** The authors have declared no competing interests exist.

## Introduction

The guava (*Psidium guajava* L.) is one of the nutritionally and commercially important fruit crops which belongs to family Myrtaceae, and the genus *Psidium* having diploid chromosome number of 2n = 22 [1]. Guava is native to tropical America and widely distributed across the cool subtropical to warm tropical countries [2, 3]. Presently, countries like India, Mexico, Brazil, Cuba, Venezuela, the USA, Australia, New Zealand and South Africa are the major guava producing countries in the world [4]. In India guava was introduced during the seventeenth century and naturalized under the tropical and subtropical parts of the country. Today, India is a leading guava producing country in the world and produces 4.05 million tonnes of guava fruits annually [5]. The area and production of guava are rapidly increasing in India due to several desirable features of the crop and currently it is the fourth important fruit crops in terms of production.

Guava is popularly known as the "Apple of the Tropics" [1], and sometimes referred to as a super fruit [6] due to its high nutritive value and antioxidant properties. The guava is also one of the choicest and palatable fruit among the consumers [7] and contains a substantial amount of vitamin A, vitamin C, niacin, riboflavin, thiamine, phosphorus, calcium, iron and edible fibre *etc.* [8, 9]. The guava fruit contains high level of polyphenolic antioxidants and ascorbic acid content which is about five times higher than oranges [10]. Owing to its high nutraceutical and medicinal values guava production can play an important role in nutritional security for the people of developing countries like India [11]. Although, the guava is a hardy crop in terms of soil and climate adaptability but it faces the various challenges like wilt disease, and deficiencies like high seed content with hard coat in major diploid commercial cultivars, while the triploid seedless varieties have small and misshapen fruits and major commercial cultivars have poor shelf-life [12, 13]. Therefore, the ideotype in guava breeding are uniform fruit shape, good fruit size, thick pulp, attractive skin and pulp colour, fewer and soft seeds, resistant to wilt disease, long storage life, dwarf stature coupled with improved yielding ability [13–16].

The varietal development in the guava through the traditional breeding is very tedious and time consuming due to its long juvenile phase and high heterozygous nature [17–19], thus yielding meagre success. Under such situations, biotechnological tools particularly genomics could apply novel techniques and strategies to solve out the existing breeding barriers in woody perennial fruit crops like guava. The molecular markers are an important genomic tool for any genetic studies among the crop species and can be adopted at each step of varietal improvement programmes starting with germplasm evaluation at cultivar or species level [20, 21], trait-specific association mapping studies [22, 23], estimation of hybridity [24, 25], construction of linkage mapping, identification of QTLs and marker-assisted selection [26]. Among the molecular markers, simple sequence repeat markers gained more importance in the studies of plant genetics due to their high reproducibility, multi-allelic in nature, co-dominant inheritance, high abundance and extensive genome coverage [27–29]. Till date very limited genomic resources particularly, microsatellite markers have been developed in guava [30–32].

The different methods have been employed for the development of the novel microsatellite markers in various crop species. The selective hybridization from the genomic libraries is considered as one of the robust, reproducible and cost-effective methods for the development of a large number of microsatellite markers in a crop species [33]. An enrichment method based on selective hybridization involves the routine steps, capturing microsatellite sequences with biotin-labelled probes that are either captured by magnetic beads coated with streptavidin or fixed on nitrate filter [34–36]. The enrichment methods based on selective hybridization have been employed for the development of the novel microsatellite markers in different crop

species including the fruit trees [37–40]. Keeping in view these facts, the microsatellite-enriched libraries were constructed to develop the novel genomic SSR loci in guava and validate those using genetic diversity and population structure analyses.

## Materials and methods

### Plant materials

A total of 40 guava genotypes comprising the wild *Psidium* species, commercial cultivars, hybrids and seedling variants were selected for the genetic diversity and population structure analyses. These genotypes are maintained and conserved in the field gene bank at Division of Fruits and Horticultural Technology, ICAR-Indian Agricultural Research Institute, Pusa Campus, New Delhi, located at geo-coordinates of 28˚04′48″N 77˚07′12″E/ 28.080˚N 77.120˚E / 28.080; 77.120. The details of selected *Psidium* genotypes are given in Table 1.

### Genomic DNA isolation

The fresh and healthy green leaves of *Psidium* genotypes were collected in icebox from the field gene bank and kept in deep freeze (- 80˚C) until further use. The CTAB method as described by Doyle and Doyle [41] was used for isolation of genomic DNA with minor modifications. 5% of polyvinylpyrrolidone (PVP) was added in the CTAB buffer to remove the polyphenols, which helped in the extraction of quality genomic DNA of the guava genotypes including the wild species. The extracted DNA was treated with the RNase (2.5 U) to remove the RNA impurities and again was purified. The quality of DNA was checked on the 0.8% agarose gel and the concentrations were confirmed with the help of spectrophotometer (Nanodrop™, Thermo-Fisher, USA). The isolated concentrated DNA was stored in the deep freeze (-80˚C) until further use.

### Construction of microsatellite-enriched libraries

The modified biotin-capture method was used to develop the microsatellite enriched libraries in *Psidium guajava* as suggested by Fischer and Bachmann [42]. The high-quality genomic DNA (1000 ng) of guava cv. Allahabad Safeda was digested with the restriction enzyme *Sau3A1* (New England Bio Labs). The restriction digestion was performed with incubation at 37˚C for overnight thereafter the restriction enzyme was inactivated by raising the temperature to 65˚C for 20 min. The success of restriction digestion was checked on the 0.8% agarose gel and smeared form of DNA indicated the success of digestion. The *Sau3A1* specific adaptors constituted with Oligo A (5′-GGC CAG AGA CCC CAA GCT TC-3′) and Oligo B (PO4—GAT CCG AAG CTT GGG GTC TCT GGC C), which were ligated with the restriction digested DNA, while using the T4 DNA ligase (Promega Corp., USA). The ligation was performed overnight at 4˚C. The success of adaptor- ligation was checked on the 0.8% agarose gel and the size ranged from 400–1000 bp was excised and eluted using the QIAquick gel extraction kit (Qiagen, Germany). Further, the success of adaptor ligation was confirmed with PCR amplification [43].

The adaptor-ligated DNA fragments were hybridized with the pre-washed (1X washing buffer and 2X washing buffer, respectively) streptavidin-coated magnetic beads and 3'-biotinylated oligonucleotides probes [(CT)$_{14}$, (GT)$_{12}$, and (AAC)$_8$], and incubated at 60˚C in 6X SSC for 30 min. with gentle agitation at each 5 min. interval and consequently, microsatellite-containing DNA fragments were obtained. The magnetic beads were removed after hybridization using the magnetic stands and then the hybridized fragments were incubated in 2X SSC and 1X SSC, respectively and subsequently, boiling at 95˚C for 15 min. in TE buffer. To obtain the

**Table 1. The details of the guava genotypes including *Psidium* species used in the study.**

| Sl. No. | Genotype | *Psidium* species | Origin/ Parentage | Major trait(s) |
|---|---|---|---|---|
| 1 | Allahabad Safeda | *P. guajava* L. | Selection | White pulp, round fruit shape |
| 2 | Allahabad Safeda (Variant) | *P. guajava* L. | Selection | White pulp, oblong fruit shape |
| 3 | Allahabad Surkha | *P. guajava* L. | Selection | Deep pink pulp, apple red exterior |
| 4 | Arka Amulya | *P. guajava* L. | Hybrid (Allahabad Safeda × Triploid) | White pulp, fruit round shape |
| 5 | Arka Kiran | *P. guajava* L. | Hybrid (Kamsari x Purple Local) | Pink pulp |
| 6 | Arka Mridula | *P. guajava* L. | Selection | White pulp |
| 7 | Arka Rashmi | *P. guajava* L. | Hybrid (Kamsari x Purple Local) | Deep pink in colour |
| 8 | Black guava | *P. guajava* L. | Selection | Deep pink pulp, purple peel, purple leaves |
| 9 | Exotic Selection | *P. guajava* L. | Selection | Waxy leaves, spreading plant |
| 10 | Guava genotype (R1 P10) | *P. guajava* L. | Selection | White pulp |
| 11 | Hisar Safeda | *P. guajava* L. | Hybrid (Allahabad Safeda x Seedless) | White pulp, upright tree growth |
| 12 | Hisar Surkha | *P. guajava* L. | Hybrid (Apple Colour x Banarasi Surkha) | Pink pulp, drooping tree |
| 13 | Hisar Surkha (Variant) | *P. guajava* L. | Hybrid (Apple Colour x Banarasi Surkha) | Pink pulp, oblong fruit shape |
| 14 | Hubsi Red | *P. guajava* L. | Selection | Red pulp |
| 15 | Kasipur Selection | *P. guajava* L. | Selection | White pulp |
| 16 | L-49 | *P. guajava* L. | Selection | White pulp |
| 17 | Lalit | *P. guajava* L. | Selection | Red pulp, suitable for table use and processing |
| 18 | Pant Prabhat | *P. guajava* L. | Selection | White pulp |
| 19 | *P. chinensis*1 | *P. chinensis* | Related species | Pink pulp, small fruits and small sunken leaves |
| 20 | *P. chinensis*2 | *P. chinensis* | Related species | White pulp, small fruits and small sunken leaves |
| 21 | *P. species* 1 | *P. species* (Unknown) | Unknown species | White pulp, small fruits and leaves |
| 22 | *P. guinense* | *P. guinense* L. | Related species | Creamy-white pulp, acidic fruits and large leaves |
| 23 | *P. quadrangularis* | *P. quadrangularis* L. | Related species | Dwarf plant, smaller fruits |
| 24 | *P. species* 2 | *P. species* (Unknown) | Unknown species | Dwarf plant, smaller fruits and bunch bearing |
| 25 | Punjab Pink | *P. guajava* L. | Portugal x L 49 = $F_1$ x Apple Colour | Pink Pulp, |
| 26 | Pusa Srijan | *P. guajava* L. | Aneuploid | Dwarf growth habit, small leaves, slow growth |
| 27 | Red Flesh Type | *P. guajava* L. | Seedling selection | Red pulp, oblong fruit shape |
| 28 | Red Variant (R3P5) | *P. guajava* L. | Seedling selection | Pink pulp, smaller fruits |
| 29 | Red Variant (R3P7) | *P. guajava* L. | Seedling selection | Pink pulp, smaller fruits |
| 30 | H-1 | *P. chinensis* x *P. guajava* L. | Open-pollinated seedling of *P. chinensis*1 | Medium size fruit, white pulp |
| 31 | H-2 | *P. chinensis* x *P. guajava* L. | -do- | Medium size fruit, white pulp |
| 32 | H-3 | *P. chinensis* x *P. guajava* L. | -do- | Medium size fruit, white pulp |
| 33 | Sasri | *P. guajava* L. | Selection | White pulp |
| 34 | Snow White | *P. guajava* L. | Selection | White pulp |
| 35 | Sweta | *P. guajava* L. | Selection | White pulp, round fruit shape |
| 36 | Thai Variant (R6P1) | *P. guajava* L. | Selection | White fruits and larger fruits |
| 37 | TN Selection | *P. guajava* L. | Selection | Pink pulp, globose fruit shape |
| 38 | Trichy | *P. guajava* L. | Selection | White pulp, yellowish-greenish skin |
| 39 | VNR Bihi | *P. guajava* L. | Selection | White pulp, larger fruits |
| 40 | Wild Type (K) | *P. species* (Unknown) | Selection | Red pulp, smaller fruits |

The fruit weight < 100 g considered as small fruits and >300 g considered as larger fruits. *P.* indicates the genus *Psidium*.

increased content of microsatellite containing DNA fragments, a PCR cycle was performed using Oligo A as a primer [43]. Thereafter, the PCR product obtained was ligated with the TA cloning vector (Invitrogen, Life Technologies) and incubated at 16°C for an hour and then the ligated vector was transformed into the competent cells of *E. coli* DH5α. The transformed

colonies were selected based on the X-gal/ IPTG selection method and the positive clones were confirmed through colony PCR, while using M13 as forward-reverse primer. The positive clones were selected and plasmids DNA were isolated using the plasmid DNA isolation kit (Zymo Research, USA) and isolated DNA were sequenced using the Sanger di-deoxy sequencing method (Sequencher Tech. Pvt. Ltd.) with M13 Primer (forward: 5'-GTAAAACGACGGC-CAGT-3').

## SSR finding and primer designing

The SSR repeats were searched from the sequences obtained from the positive clones, using the SSR finder software (http://www.csufresno.edu/ssrfinder/). The Primer 3.0 software version 0.4.0 (http://bioinfo.ut.ee/primer3-0.4.0/) was used to design the novel g-SSR primers. The obtained sequences having the nucleotide ranged from 18 to 25 with the amplicon size ranged from 100 to 500 bp were used for the primer designing.

## PCR amplification and gel-electrophoresis

A total of 38 g-SSR primers were designed, synthesized and selected for the genetic diversity, population structure and cross-transferability studies among 40 *Psidium* genotypes including the wild *Psidium* species. Out of 38 g-SSRs, 26 gave polymorphic amplicons among the selected set of guava genotypes. The genomic DNA of all the *Psidium* genotypes was isolated from the new leaf flushes and the final working concentration was maintained at 20 ng/µl. The thermocycle was performed with reaction mixture volume of 16 µl containing 2.5 µl genomic DNA (50 ng), 1.5 µl of 10X buffer, 2 µl of 2.5 mM $MgCl_2$, 1.5 µl of 10 mM dNTPs, 0.6 µl of each primer (10 nmol), 0.4 µl of *Taq* DNA polymerase enzyme (Thermo Scientific, USA) and final volume was made with addition of nuclease free double-distilled water. The following PCR conditions were maintained during the amplification of each developed SSR primer: Initial denaturation temperature was kept at 94˚C for 5 min followed by 36 cycles of denaturation at 94˚C for 30 sec at annealing temperature (temperature was standardized by gradient PCR for each SSR primer), 72˚C for 1 min maintained for extension and 72˚C for 10 min for final extension. The amplified product of each SSR was separated on 4% metaphor agarose (Lonza, USA) gel for 4 h at 100 V and gel images were captured in gel documentation system (Alpha Imager®, USA).

## Data scoring and statistical analyses

The scoring of amplified PCR products for each primer among the selected 40 *Psidium* genotypes was done using the software PyElph 1.4 [44]. Genetic diversity parameters, *viz*., the major allele frequency ($M_{af}$), Allele number ($A_n$), gene diversity (GD) or expected heterozygosity, observed heterozygosity ($H_o$) and the polymorphic information content (PIC) for each g-SSR was calculated using, Power Marker 3.5 [45]. The allele frequency of each SSR locus was also calculated using the allele frequency procedure in Power Marker 3.5. The allele having frequency <0.05 considered as rare allele and allele amplified only in the one genotype referred as unique allele [46, 47]. In addition, genetic distances among the all selected *Psidium* genotypes and Neighbour-joining (NJ) tree were constructed using Power Marker 3.5 [45]. The population structure of the *Psidium* was constructed using a model-based clustering algorithm implemented in the STRUCTURE software version 2.3.4 [48]. The model-based Bayesian algorithm was applied to identify the genetically unique sub-populations of the *Psidium* genotypes based on the allele frequencies of developed g-SSRs. Initially, the value of K was set 2–10 to identify the genetic cluster of the *Psidium* genotypes. ΔK values are derived after plotting Ln (PD) and ΔK value was used to identify the appropriate numbers of the genetic cluster [49].

An online available programme- Structure Harvest was used to obtain the final population structure of the *Psidium* genotypes (http://taylor0.biology.ucla.edu). The analysis of molecular variance (AMOVA), Pair-wise genetic distances and principal coordinate analysis (PCoA) were performed using the software GenAlEx 6.5 [50].

## Results

### Development of microsatellite enriched libraries and development of novel g-SSR markers

The high-quality genomic DNA of *Psidium guajava* cv. Allahabad Safeda was used for the construction of microsatellite libraries, which were enriched with the three types of 3'-biotinylated oligonucleotide probes, *viz.*, $(CT)_{14}$, $(GT)_{12}$, and $(AAC)_8$. The recombinant clones were randomly selected from the transformed colonies of *E. coli* (DH5α) and further the positive clones were screened through colony PCR. A total of 153 transformed colonies were randomly selected of which 111 were found positive and subsequently, subjected to Sanger sequencing (Table 2). Furthermore, sequences of the recombinant clones were screened for the presence of the enriched microsatellite repeats (CT, GT, and AAC), out of which only 33 clones had the desired microsatellite sequence repeats and was confirmed through SSR Finder software. It was also recorded that some of the positive clones harbored more than one SSR repeats. The microsatellite regions having more than five motif repeats were selected for primer designing. A total of 38 novel g-SSRs primers were designed from 33 positive recombinant clones and synthesized for the diversity, population structure and transferability analyses (Table 3). The designed g-SSRs contained the motif groups from monomer to pentamer. The percentage of their occurrence varied from 2.63 to 89.47. The dimer motif group occurrence was 89.47%, while other motif groups occurred with frequencies of 2.63% (Table 4).

### Genetic diversity parameters for g-SSRs markers

Out of the 38-novel g-SSRs markers tested, 26 gave polymorphic amplicons and showed substantial genetic diversity among the studied guava genotypes including wild *Psidium* species (Table 5 & Fig 1). The amplified allelic size ranged from 140 to 550 bp among 40 *Psidium* genotypes with 26-SSR loci. A total of 90 alleles were amplified among 26 g-SSRs markers, which varied from 2 to 11 alleles with an average of 3.46 per locus. The major allelic frequency varied from the 0.17 to 0.94 among the g-SSRs with an average value of 0.56 per locus. The SSR, GUV4-50 had the highest allelic frequency (0.94), while SSR GUV2-43 had the lowest value (0.17). The gene diversity of the g-SSRs was calculated among 26 g-SSR loci, which ranged from 0.11 to 0.88 with an average of 0.53 per locus. The SSR GUV2-43 had the highest gene diversity (0.88), while the lowest value (0.53) was recorded in SSR GUV4-50. The observed heterozygosity among the g-SSRs was also estimated, which varied from 0.00 to 0.97 with an average value of 0.29 per locus. The polymorphic information content (PIC) of the

**Table 2. Summary of genomic microsatellite markers development.**

| Sl.No. | Step | Total No./ percentage |
|---|---|---|
| 1. | Number of positive clones selected after transformation | 153 |
| 2. | Number of positive clones screened with colony PCR | 111 |
| 3. | Percentage of positive clones selected after colony PCR | 72.55% |
| 4. | Number of positive clones subjected to sanger sequencing | 111 |
| 5. | Number clones containing SSR motif | 33 |
| 6. | Percentage of clones containing SSR motif | 30% |

**Table 3. The details of motif length, product size and sequence of developed g-SSR primers.**

| Sl. No. | Primer–ID | Repeat motif | Product size (bp) | FW-Primer Sequence | RV- Primer Sequence |
|---|---|---|---|---|---|
| 1 | GUV3 | (TC)13 | 233 | AGCTTCGGATCGTAGGGTTC | TCTGTATGTGTGCTCAACCTCC |
| 2 | GUV13-2 | (AG)6 | 374 | AGCTTCGGATCAGTTAGTCCCT | CAGCTCTCTCTCAGCCTCTCTC |
| 3 | GUV19 | (AG)18 | 374 | AGCGAGGTATTGGTGAGATAGC | GTTTCTGACTTTTCACGTTCCC |
| 4 | GUV38-2 | (TC)14 | 159 | CCTTCAAATGCTCTCCTTCCTA | CTCTTCATCGTCTTCCTTCCTG |
| 5 | GUV44 | (TC)24 | 239 | CTGGCATTTAGGTCTTACGCA | ACGGTGATGATGGATGAACATA |
| 6 | GUV45-2 | (GA)6 | 206 | AGCTTCGGATCAGTTAGTCCCT | TCTCGCAGACCTCTCTCTCTCT |
| 7 | GUV25-2 | (C)10 | 286 | CACGAAAAGGCAAGAAACCTAA | CCACGGAAATGTGAAGTCCTAT |
| 8 | GUV26-1 | (GA)8 | 272 | TTGATGCCCAAAGAGAAATAGC | CAGTCCCTCTCGCAGCTC |
| 9 | GUV31 | (GA)23 | 274 | GTGTTGGAGAGGTTTTGTGTGA | TTGCGGTACATGGTTTCTTATG |
| 10 | GUV36-2 | (AG)6 | 281 | AGCTTCGGATCAGTTAGTCCCT | TCTCTCTCTCCCTATCTCGCAC |
| 11 | GUV32 | (AG)6 | 283 | AGCTTCGGATCAGTTAGTCCCT | TCCCAGTCTCTCTCTATCTCGC |
| 12 | GUV32-2 | (GA)9 | 204 | AGCTTCGGATCAGTTAGTCCCT | TCTCGCAGACCTCTCTCTCTCT |
| 13 | GUV1-37 | (CT)8 | 369 | TCCCAGTCTCTCTCTATCTCGC | AGCTTCGGATCAGTTAGTCCCT |
| 14 | GUV1-38-1 | (CTCTG)4 | 159 | CCTTCAAATGCTCTCCTTCCTA | CTCTTCATCGTCTTCCTTCCTG |
| 15 | GUV4-10 | (AG)6 | 293 | CTCTCATCATCGCTTCCAATC | TCTCTCTCGCAGGTCTCTCTCT |
| 16 | GUV4-33 | (GA)7 | 226 | AGCTTCGGATCAGTTAGTCCCT | CTCCCTCTCTCCCAGTCTCTCT |
| 17 | GUV2-17 | (AG)7 | 393 | GCTTCGGATCAGTTAGTCCCTT | CTCTCTCTCGCAGCTCTCTCTC |
| 18 | GUV2-17-2 | (GA)7 | 271 | AGCGATAGTGCGAGATAGGG | TACAGTCTCCCTTGCAGCG |
| 19 | GUV2-19 | (GA)7 | 354 | CTTCGGGTCAGTTAGTCCCTT | CTCTCTCCATCTCGCACTCTCT |
| 20 | GUV2-19-2 | (GA)7 | 270 | GAGCGATAGTGCGAGATAGGG | GTTACAGTCTCCCTTGCAGCTC |
| 21 | GUV2-21 | (AG)6 | 281 | AGCTTCGGATCAGTTAGTCCCT | TCTCTCTCTCCCTATCTCGCAC |
| 22 | GUV2-22 | (AG)6 | 387 | GCTTCGGATCAGTTAGTCCCTT | CTCTCTCTCGCAGCTCTCTCTC |
| 23 | GUV2-22-2 | (GA)7 | 477 | AGCTTCGGATCAGTTAGTCCCT | TATCTCGCACTCTCTCTCTGGC |
| 24 | GUV2-36 | (TC)20 | 320 | CTCTCTCTCTCTCTCCACAGCC | ATGACCCTCAACATAGCGTTTT |
| 25 | GUV2-36-2 | (TC)14 | 389 | AGCTTCGGATCTCTCTCCACT | ATGACCCTCAACATAGCGTTTT |
| 26 | GUV2-37 | (GA)11 | 229 | TCTGTATGTGTGCTCAACCTCC | AGCTTCGGATCGTAGGGTTT |
| 27 | GUV2-41 | (AG)6 | 325 | TGAGAGAGAGACAGATTGCGAG | CAGCCCTCTTTGGTAAGCTCT |
| 28 | GUV2-43 | (GA)9 | 228 | CCAAGAAGAGAGAAAGAGACGG | ATGTATGGTGAGTGTGTGAGGG |
| 29 | GUV4-33 | (GA)6 | 226 | AGCTTCGGATCAGTTAGTCCCT | CTCCCTCTCTCCCAGTCTCTCT |
| 30 | GUV4-34 | (AG)6 | 240 | AGCTTCGGATCAGTTAGTCCCT | TTTTCTCCCAATTCTCTTCCAG |
| 31 | GUV4-42 | (CT)21 | 301 | GTCTTAACGGCCCCTTTTGT | CAAGCTTCGGATCGAGAGAG |
| 32 | GUV4-42-2 | (TCAA)3 | 287 | CCCCTTTTGTTTTCACCTAACA | CTTCGGATCGAGAGAGAGAGAG |
| 33 | GUV4-45 | (AG)20 | 357 | CTACACCAGCAAGAACCAAACA | GTTTCTGACTTTTCACGTTCCC |
| 34 | GUV4-50 | (TC)17 | 383 | CTCTCTCTCTCTCTCCACAGCC | GAAAAGCGGGATTGATTTACAG |
| 35 | GUV4-50-2 | (TC)12 | 448 | AGCTTCGGATCTCTCTCCACT | GAAAAGCGGGATTGATTTACAG |
| 36 | GUV4-53 | (GCC)6 | 238 | TATCCCTCATCTCCTTCACCTT | ATATTAGAGAGCCTGTGGTCCG |
| 37 | GUV4-55 | (AG)6 | 388 | GCTTCGGATCAGTTAGTCCCTT | AGCTCTCTCTCGCAGCTCTCT |
| 38 | GUV4-55-2 | (GA)7 | 476 | AGCTTCGGATCAGTTAGTCCCT | TCTATCTCGCACTCTCTCTGGC |

**Table 4. Details of occurrence of motif length group during the genomic microsatellite development.**

| Sl.No. | Motif length group | Total no. | Percentage |
|---|---|---|---|
| 1 | monomer | 1 | 2.63 |
| 2 | dimer | 34 | 89.47 |
| 3 | trimer | 1 | 2.63 |
| 4 | tetramer | 1 | 2.63 |
| 5 | pentamer | 1 | 2.63 |

**Table 5. Genetic variability indices of the g-SSRs among the set of guava genotypes.**

| Sl. No. | Marker ID | Annealing temp. (Ta) | Allele size (bp) | $M_{af}$ | $A_n$ | GD | Ho | PIC |
|---|---|---|---|---|---|---|---|---|
| 1. | GUV-3 | 45.9 | 140–160 | 0.64 | 2.00 | 0.46 | 0.00 | 0.36 |
| 2. | GUV13-2 | 62.6 | 180–200 | 0.65 | 2.00 | 0.46 | 0.00 | 0.35 |
| 3. | GUV-19 | 63.3 | 320–420 | 0.34 | 6.00 | 0.74 | 0.03 | 0.70 |
| 4. | GUV38-2 | 60 | 150–170 | 0.86 | 2.00 | 0.24 | 0.00 | 0.21 |
| 5. | GUV45-2 | 63.9 | 210–280 | 0.56 | 2.00 | 0.49 | 0.89 | 0.37 |
| 6. | GUV26-1 | 57.1 | 280–360 | 0.52 | 2.00 | 0.50 | 0.96 | 0.37 |
| 7. | GUV31 | 60 | 230–280 | 0.39 | 6.00 | 0.77 | 0.39 | 0.74 |
| 8. | GUV36-2 | 60 | 220–300 | 0.43 | 4.00 | 0.64 | 0.80 | 0.57 |
| 9. | GUV32 | 43 | 200–280 | 0.31 | 6.00 | 0.78 | 0.00 | 0.75 |
| 10. | GUV32-2 | 43 | 170–185 | 0.67 | 2.00 | 0.44 | 0.00 | 0.35 |
| 11. | GUV1-37 | 40.9 | 200–420 | 0.26 | 8.00 | 0.79 | 0.83 | 0.75 |
| 12. | GUV 1-38-1 | 40.9 | 150–175 | 0.75 | 2.00 | 0.38 | 0.50 | 0.30 |
| 13. | GUV4-10 | 46.9 | 380–520 | 0.51 | 3.00 | 0.61 | 0.97 | 0.54 |
| 14. | GUV4-33 | 46.9 | 160–200 | 0.75 | 2.00 | 0.38 | 0.25 | 0.30 |
| 15. | GUV2-19 | 50.9 | 190–210 | 0.55 | 2.00 | 0.50 | 0.16 | 0.37 |
| 16. | GUV2-21 | 56.9 | 230–260 | 0.67 | 2.00 | 0.44 | 0.19 | 0.34 |
| 17. | GUV2-36-2 | 56.9 | 400–550 | 0.92 | 2.00 | 0.15 | 0.16 | 0.14 |
| 18. | GUV2-37 | 56.9 | 240–270 | 0.71 | 2.00 | 0.41 | 0.00 | 0.32 |
| 19. | GUV2-43 | 57.6 | 170–390 | 0.17 | 11.00 | 0.88 | 0.00 | 0.87 |
| 20. | GUV4-33 | 60 | 220–230 | 0.81 | 2.00 | 0.30 | 0.00 | 0.26 |
| 21. | GUV4-42 | 50.9 | 420–470 | 0.55 | 2.00 | 0.50 | 0.91 | 0.37 |
| 22. | GUV4-42-2 | 50.9 | 200–260 | 0.40 | 4.00 | 0.68 | 0.00 | 0.62 |
| 23. | GUV4-45 | 50.9 | 280–380 | 0.55 | 4.00 | 0.61 | 0.00 | 0.55 |
| 24. | GUV4-50 | 56.9 | 190–320 | 0.94 | 2.00 | 0.11 | 0.12 | 0.10 |
| 25. | GUV4-50-2 | 56.9 | 250–400 | 0.38 | 4.00 | 0.72 | 0.47 | 0.68 |
| 26. | GUV4-53 | 56.9 | 200–260 | 0.40 | 4.00 | 0.68 | 0.00 | 0.62 |
| | **Mean** | | | **0.56** | **3.46** | **0.53** | **0.29** | **0.46** |

**Where:** Maf = major allele frequency, An = Allele number, GD = gene diversity, Ho = observed heterozygosity, PIC = polymorphic information content.

developed SSR was also calculated, which ranged from 0.10 (SSR GUV4-50) to 0.87 (SSR GUV2-43) with an average value of 0.46 per locus. The 11 g-SSRs, *viz.*, GUV-19, GUV31, GUV36-2, GUV32, GUV1-37, GUV4-10, GUV2-43, GUV4-42-2, GUV4-45, GUV4-50-2 and GUV4-53 had the PIC value >0.5 for the guava genotypes, which indicated their high discrimination power. Furthermore, out of 90 alleles amplified, 7 alleles had the frequency less than 0.05, thus they could be considered as the rare alleles, while 3 alleles were identified as unique alleles in *P. chinensis* 2, Wild type (k) and Arka Kiran.

## Phylogenetic and cross-transferability studies

The Neighbor-joining (N-J) tree was constructed using the data scored from 26 polymorphic g-SSRs, which genetically differentiated the *Psidium* genotypes including the wild species into two major clusters (Fig 2). The genotypes, namely, R1P10, Hisar Safeda, Hisar Surkha and Hybrid 3 grouped into cluster 1, while remaining genotypes including the wild guava species grouped into cluster 2. The N-J tree was further simplified into six clades with two out-groups for its better explanation and understanding. Clade 1 contained four guava genotypes, *viz.*, R1P10, Hisar Safeda, Hisar Surkha and Hybrid 3. The Clade 2 was formed by Allahabad Surkha, Arka Amulaya, Arka Kiran, Allahabad Safeda, Allahabad Safeda (variant), Hybrid 1, Red

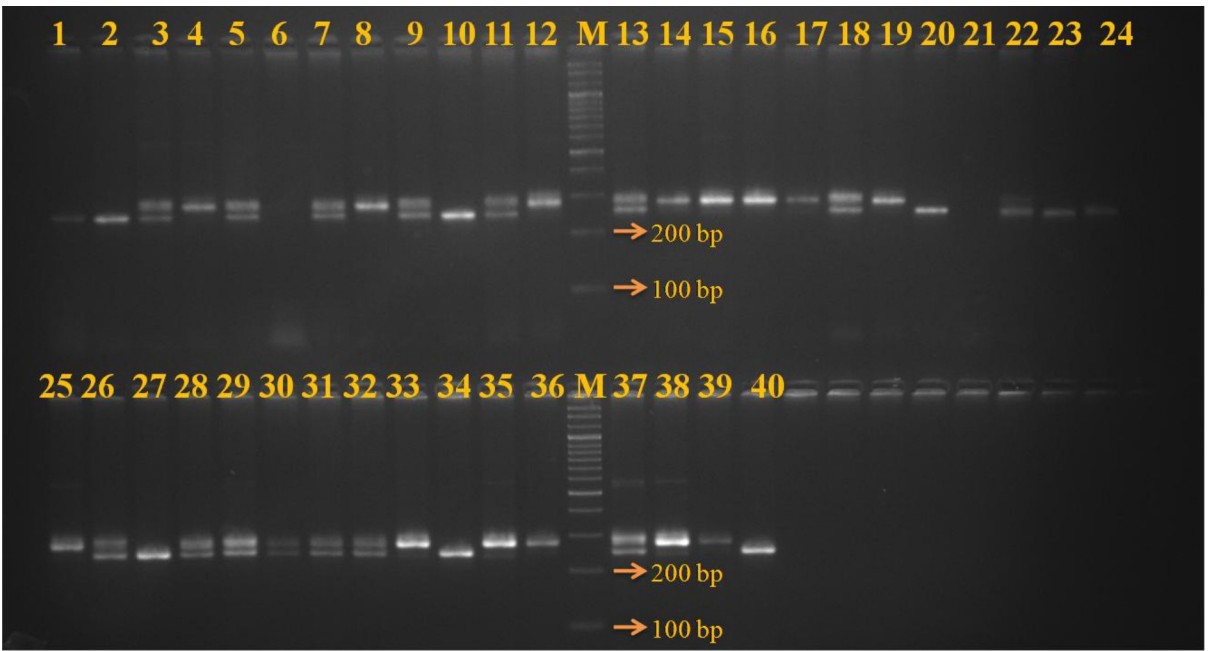

**Fig 1. Gel image of SSR locus GUV31 showing allelic variation among the *Psidium* genotypes.**

Type (Variant R3P5), Punjab Pink and Red type (Variant R3P7). The Clade 3 constituted by the guava genotypes, *viz*., Arka Rashmi, Black guava, Exotic Selection, Red pulped genotype, Arka Mridula and Pusa Srijan, while clade 4 contained only two genotypes, *viz*., Sasri and Sweta. Clade 5 contained most of the selected wild *Psidium* species, *viz*., *P. guinense*, *Psidium* sp. 2, *P. quadrangularis*, *P. chinensis* 2, and *P.* sp. 1. Clade 6 constituted by 12 guava genotypes, *viz*., TN Selection, Wild type (K), Trichy, Hisar Surkha (Variant), Hubsi Red, *P. chinensis* 1, VNR Bihi, Pant Prabhat, Thai Variant, Kasipur Selection, L-49 and Lalit. Besides, the clades, cluster also had two out groups. The guava genotype Hybrid 2 was in the out-group 1, while genotype Snow White was in out-group 2. The minimum genetic distance (0.047) was recorded between the guava genotypes L-49 and Lalit, while, maximum genetic distance (0.899) was recorded between *Psidium* species 2 & guava genotype (R1P10); and *Psidium* species 2 & L-49. Moderate to high cross-transferability of the developed g-SSRs was recorded among the studied wild *Psidium* species, which varied from 38.46 (*P.* species 2) to 80.77% (*P.* species 1 & *P. chinensis* 2). With regard to SSRs amplification in wild *Psidium chinensis*1, 20 SSRs were amplified, while 6 SSRs (GUV26-1, GUV32, GUV32-2, GUV2-37, GUV2-43 and GUV4-50-2) did not yield any amplicon. In *P. chinensis* 2, 21 SSRs yielded amplicons, while 5 (GUV45-2, GUV26-1, GUV32-2, GUV2-37 and GUV4-50-2) could not produce any amplicon. Similarly, in another unknown species named as *Psidium* sp.1, 21 SSRs were amplified, while 5 (GUV31, GUV36-2, GUV32-2, GUV2-21 and GUV2-37) did not give any amplification. In, *P. guinense*, 17 SSRs gave amplicons, while 9 (GUV-3, GUV32, GUV32-2, GUV4-10, GUV4-33, GUV2-21, GUV2-36-2, GUV2-37 and GUV4-50-2) did not give any amplicon. In another wild species *P. quadrangularis*, 12 SSRs (GUV13-2, GUV-19, GUV38-2, GUV45-2, GUV26-1, GUV31, GUV 1-38-1, GUV2-43, GUV4-42-2, GUV4-45, GUV4-50 and GUV4-53) were amplified while 14 did not yield any amplicon. Furthermore, in *P.* species 2, only 10 SSRs (GUV-3, GUV13-2, GUV-19, GUV38-2, GUV45-2, GUV31, GUV 1-38-1, GUV4-42-2, GUV4-45 and GUV4-53) yielded amplicon, while 16 did not produce amplicon. In another wild type guava genotype, 11 SSRs (GUV-19, GUV31, GUV36-2, GUV1-37, GUV 1-38-1,

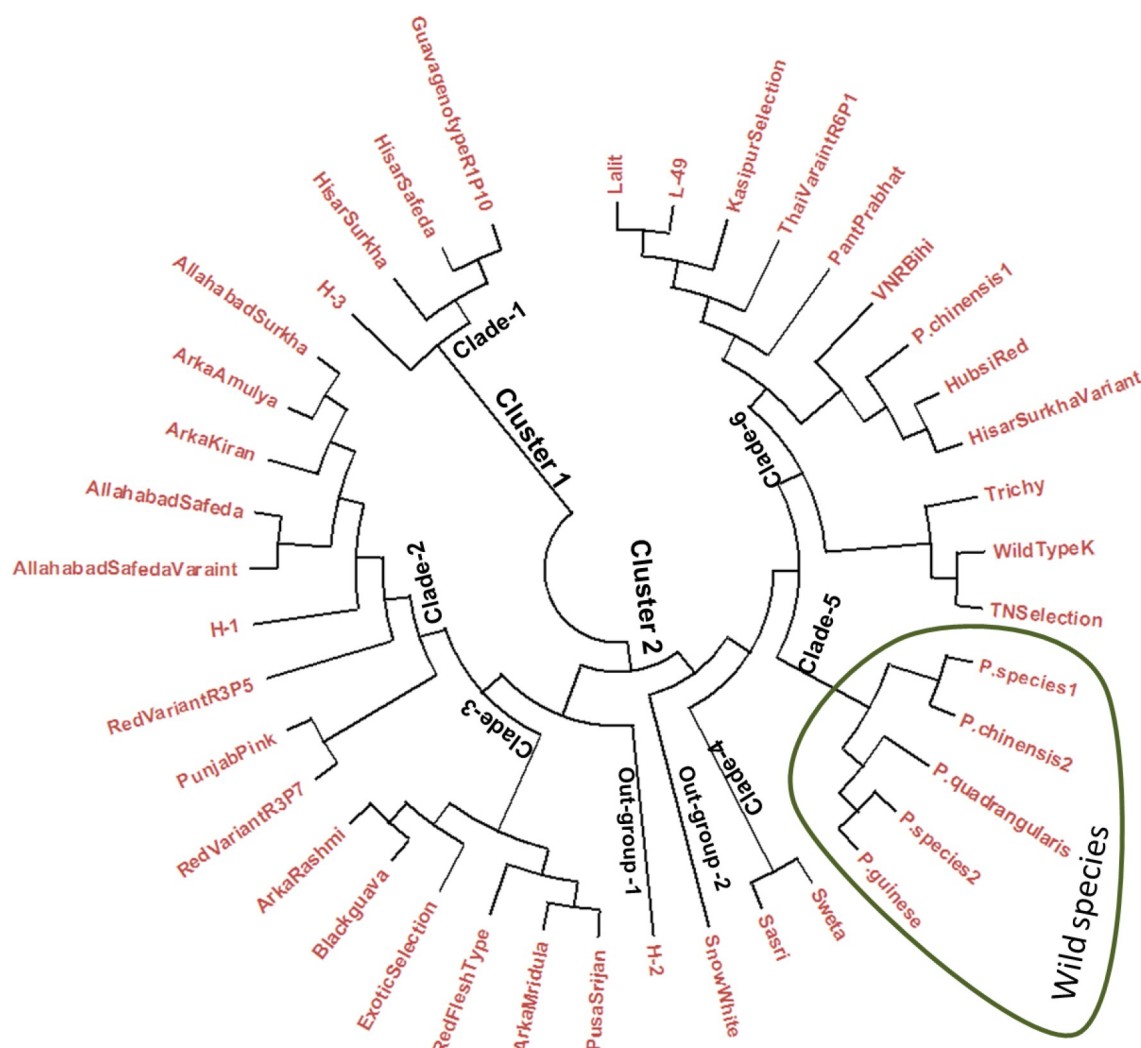

**Fig 2. N-J tree of guava genotypes including wild *Psidium* species using scored data of the 26 developed g-SSRs.**

GUV4-10, GUV4-33, GUV2-19, GUV2-21, GUV2-36-2, GUV2-43, GUV4-42, GUV4-42-2, GUV4-45 and GUV4-53) amplified, while 15 did not yield any amplicon. Only two SSR, *viz.*, GUV32-2 and GUV2-37 out of 26 polymorphic SSRs did not produced any amplicon among the selected wild *Psidium* species.

## Population structure analysis

The studied guava genotypes including the wild *Psidium* species were grouped into three genetic population groups and recorded the most probable number of K (Fig 3) and grouping of guava individuals were illustrated using bar plot diagram (Fig 4). The population I contained five genotypes, *viz.*, Allahabad Surkha, Arka Amulya, Allahabad Safeda (Variant), Allahabad Safeda and Arka Kiran. The population II constituted by 17 guava genotypes, *viz.*, *P. chinensis* 2, *P. chinensis*1, Lalit, *P. guinense*, L-49, Pant Prabhat, Hubsi Red, Kasipur Selection, *P. quadrangularis*, *P.* species 2, VNR Bihi, *P.* species 1, Wild Type (K), Thai variant (R6P1), Trichy, Hisar Surkha (variant), and TN Selection, which contained all the studied wild *Psidium* species. Subsequently, the population III comprised of 18 guava genotypes, *viz.* Hisar Safeda,

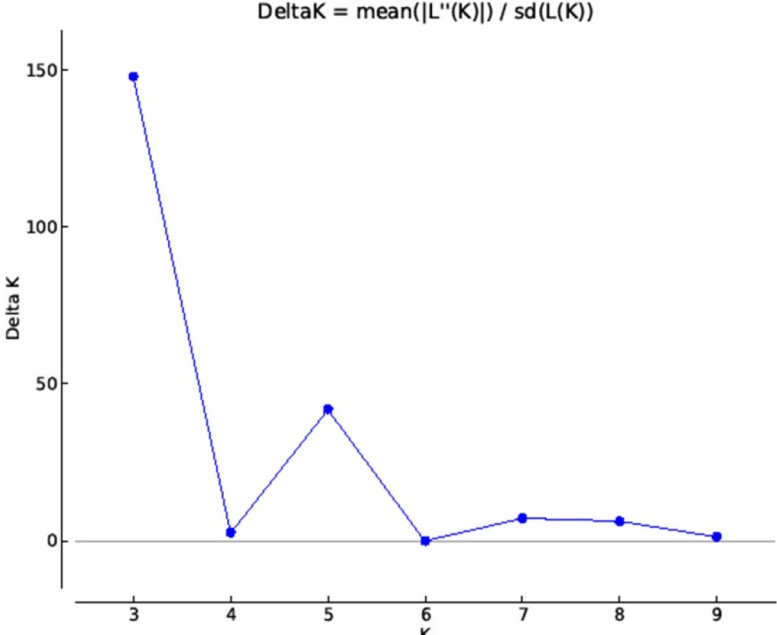

**Fig 3. Estimation of *Psidium* population using LnP(D) derived Δk with k ranged from 2 to 10.**

guava genotype R1P10, H-2, H-3, Punjab Pink, Pusa Srijan, Red Flesh Type, Red Variant (R3P7), Hisar Surkha, Sweta, Arka Mridula, Sasri, Black guava, Red Variant (R3P5), Exotic Selection, Arka Rashmi, Snow White, H-1. The Fsts values of the guava populations were also estimated, the mean Fst values of the population I, population II and population III were 0.4434, 0.3275 and 0.3228, respectively, while the mean alpha value was 0.0603 (Table 6). The allele-frequency divergence between population I and population II was 0.2961, between population I and population III was 0.1539; and population II and population III was 0.1925 (Table 7).

## AMOVA and PCoA

The analysis of molecular variance (AMOVA) of three guava populations (as differentiated by model-based population structure analysis) explained 19% variations among the populations;

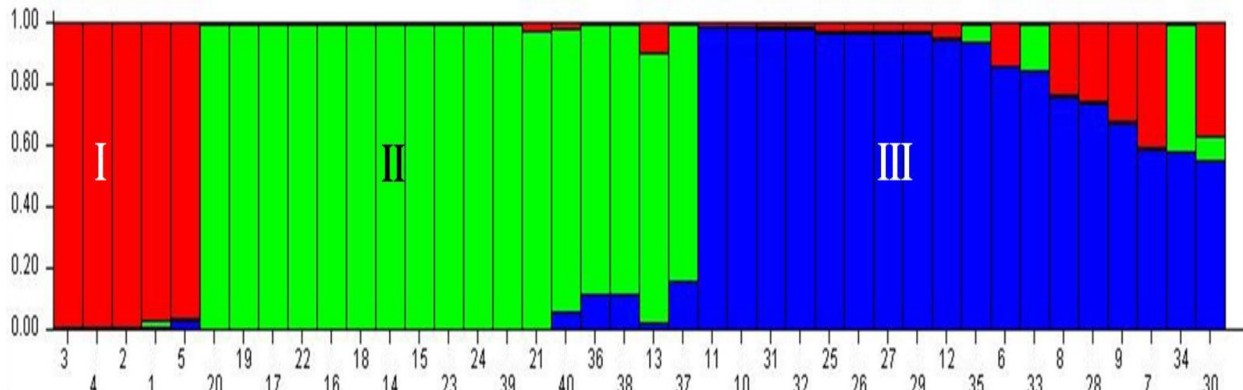

**Fig 4. Barplot of population structure (K = 3) of 40 *Psidium* genotypes based on 26 g-SSRs.** The serial number of the guava genotypes in the barplot follows Table 1.

**Table 6. Mean value of Fst1, Fst2, Fst3 and alpha inferred from model-based approach.**

| | |
|---|---|
| Mean value of Fst_1 | 0.4434 |
| Mean value of Fst_2 | 0.3275 |
| Mean value of Fst_3 | 0.3228 |
| Mean value of alpha | 0.0603 |

50% variations among the individuals and 31% within the individuals (Fig 5 and Table 8). Further, principal coordinate analysis (PCoA) revealed significant high genetic variation among the studied guava genotypes. The three axes of PCoA explained 37.46% cumulative variation wherein first, second and third axes explained 16.71, 12.21 and 8.54% of genetic variation, respectively (Table 9). In the PCoA, the different population groups denoted by three different colours, which distributed over the coordinates and closer genetic relationship among the studied wild *Psidium* species were also illustrated (Fig 6).

## Discussion

### Microsatellite markers development

Microsatellite or simple sequence repeats (SSRs) are widely distributed in the eukaryotes genomes. These microsatellites are used as molecular markers and have various applications in genetics and plant breeding studies. Thus, the availability of microsatellite markers in crop species of interest is essential for conducting the genetic studies and facilitates the crop improvement programme; and guava is one of the commercial important crop, which essentially required microsatellite markers. Various techniques have been utilized for the generation of microsatellite markers in different crop species including the fruit crops of which constructions of the genomic libraries and screening of the positive clones; and subsequent sequencing is considered one of the effective methods [51]. The development of enriched genomic libraries reduces the cost of the microsatellite markers and enhances the efficiency of marker development [35, 36]. Thus, the microsatellite library enrichment techniques have been used for the development of microsatellite markers in various crop species including the fruit species, date palm [40], peach [52–54], lychee [55], mango [56], Chinese jujube [57], and recently in *Bauhinia strychnifolia* [58]. In the present investigation, genomic libraries were constructed using the genomic DNA of guava cv. Allahabad Safeda and enriched with three microsatellite repeats, *viz.*, $(CT)_{14}$, $(GT)_{12}$ and $(AAC)_8$. A total of 111 clones were found positive and subjected for Sanger sequencing, out of which 33 clones were with the desired microsatellite repeats. However, the success rate of microsatellite markers from clones (30%) was lesser than previously reported by Duval et al. [56] in mango and Al-Faifi et al. [40] in date palm. It was also noticed that few of the clones harbored more than one microsatellite markers, thus a total of 38 SSRs could be developed. The maximum occurrence of dimer motif repeats was recorded among the developed genomic SSR markers. High frequency occurrence of dimer motif was also reported in other plant species, *viz.*, date palm [40], rubber tree [59], tea [60], istachio nut [61] and *Melilotus* [62]. This is the second report on development of microsatellite markers in

**Table 7. Allele-frequency divergence among populations of guava genotypes.**

| | Pop 1 | Pop 2 | Pop 3 |
|---|---|---|---|
| Pop 1 | - | 0.2961 | 0.1539 |
| Pop 2 | 0.2961 | - | 0.1925 |
| Pop 3 | 0.1539 | 0.1925 | - |

## Percentages of Molecular Variance

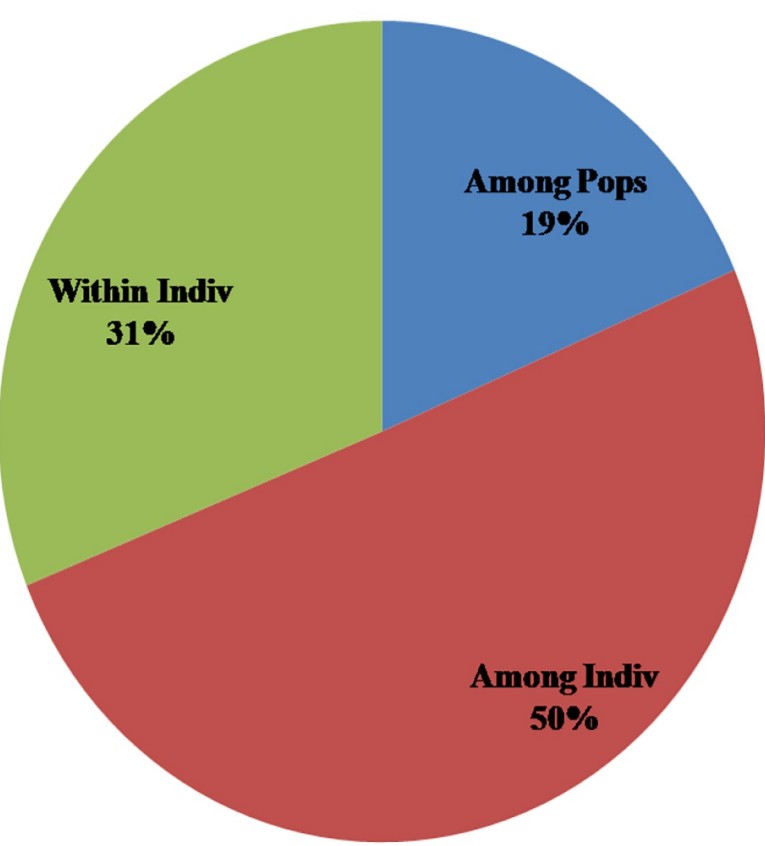

**Fig 5. Analysis of molecular variance (AMOVA) of 40 *Psidium* genotypes based on 26 g-SSRs.**

guava through the enriched library technique. Earlier, Risterucci et al. [30] designated and synthesized 23 novel microsatellite markers using library enrichment technique. The expected amplicon size ranged from 300 to 500 bp among the developed SSR markers, after trimming off the vector sequences.

### Genetic variability indices

The genetic variability indices are important measure to assess the informativeness of the developed genomic SSR markers. The 38 novel genomic SSRs were developed out of which 26 SSR were found polymorphic among the selected guava genotypes and *Psidium* species. The

**Table 8. Analysis of molecular variance of 26 g-SSRs among 40 guava genotypes.**

| Source | df | SS | MS | Est. Var. | % |
|---|---|---|---|---|---|
| Among populations | 2 | 108.257 | 54.128 | 1.738 | 19 |
| Among individuals | 37 | 456.293 | 12.332 | 4.710 | 50 |
| Within individuals | 40 | 116.500 | 2.913 | 2.913 | 31 |
| Total | 79 | 681.050 | | 9.360 | 100 |

**Table 9. Percentage of variation explained by the first 3 axes among the guava genotypes.**

| Axis | 1 | 2 | 3 |
|---|---|---|---|
| Variation (%) | 16.71 | 12.21 | 8.54 |
| Cumulative variation (%) | 16.71 | 28.92 | 37.46 |

genetic variability indices of each developed SSR locus was calculated and an average value of allele number, major allele frequency, gene diversity, observed heterozygosity and polymorphic information of each developed SSR locus were 3.46, 0.56, 0.53, 0.29 and 0.46, respectively. In the past, various fruit crop species including the guava germplasm were characterized using the SSR markers and genetic variability indices were also estimated that indicate their suitability for genetic diversity studies. For example, Tuler et al. [63] calculated an average of expected heterozygosis, observed heterozygosis and PIC value of 31 SSR loci was 0.47, 0.15 and 0.43, respectively among 13 studied *Psidium* species. Further, Al-Faifi et al. [40] studied the diversity indices of 22 SSRs among the 32 date palm genotype and they calculated the average PIC value was 0.595. Kherwar et al. [64] illustrated the generic variability parameters of 24 SSRs, while conducting diversity study on 36 guava genotypes where they obtained average allele number, major allele frequency, gene diversity, PIC values as 3.682, 0.561 0.548, 0.490, respectively. Recently, Ma et al. [65] estimated genetic diversity indices of 15 polymorphic SSRs on 45 guava accessions and recorded, average allele numbers per locus, PIC value as 4.3 and 0.60, respectively. The diversity indices of the developed SSRs were found congruent with the SSS loci used in the above-cited studies. Thus, 26 the developed novel g-SSRs were showed to have substantial genetic variability indices among the studied guava genotypes and *Psidium* species. Furthermore, SSR markers are considered more informative or high discrimination power when microsatellite markers have the polymorphic information content value ≥0.5 [66–68]. In the present investigation, 11 developed SSR loci had the PIC value > 0.5, which indicates their high discrimination power. These SSRs could be efficiently utilized in hybridity analysis, assessment of genetic variability, fingerprinting and identifications of the crop varieties [69]. Thus, the developed genomic SSRs, *viz*., GUV-19, GUV31, GUV36-2, GUV32, GUV1-37, GUV4-10, GUV2-43, GUV4-42-2, GUV4-45, GUV4-50-2 and GUV4-53 could be the most appropriate to include in any genetic studies in guava.

## Cross-transferability study

The development of SSR markers involved time, labour and cost, which necessitate the cross-transferability studies of developed markers among the related species and genera, where the genome sequence information or genetic map is lacking [63, 70]. The sequence information of several crop species revealed that closely related species/ genera have more homology in their flanking region of the SSR loci, thus they share common genomic region [28, 71–73]. It was observed that the SSR loci developed in one crop species may show a high cross-transferability percentage among the closely related genera or species. Thus, the SSRs obtained from the one species could be used for the phylogenetic studies and comparative genome mapping in related species and genera [63, 74]. In the present investigation, inter-specific cross-transferability of the developed markers varied from 38.46 to 80.77% among the selected wild *Psidium* species. Peakall et al. [75] also suggested that the cross-transferability of SSRs among the species of the same genus can differ from 50 to 100%. Similarly, Tuler et al. [63] and Sitther et al. [76] recorded the high cross-transferability of SSRs among the other species of the genus *Psidium*. Recently, Falahati-Anbaran et al. [77] also reported a high rate of cross-species transferability of the developed SSR markers in *Dalechampia scandens*, while Thakur et al. [78] too obtained

**Principal Coordinates (PCoA)**

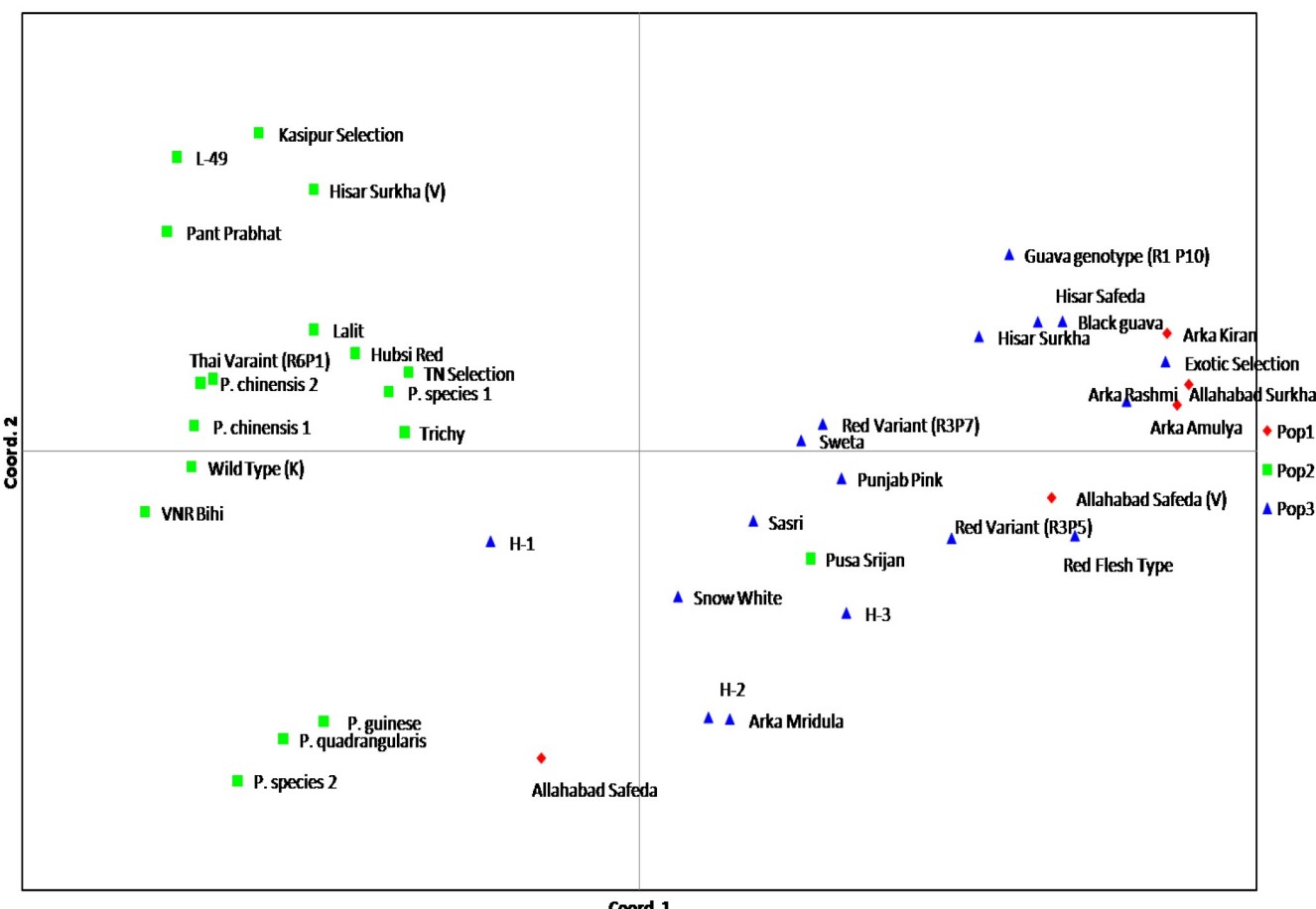

**Fig 6. Principal coordinate analysis (PCoA) of 40 *Psidium* genotypes based on 26 g-SSRs.**

100% cross-transferability of *Brassica* derived SSR for *B. juncea* and subspecies of *B. rapa*. In the present, investigation, the moderate to high transferability percentage of the developed SSRs confirmed that the studied *Psidium* species have more evolutionary closeness with cultivated guava species (*P. guajava* L.). The high cross-species transferability of the developed genomic SSR loci can be used in comparative genomics, conservation genetics and other varietal improvement programme of guava comprising with the wild *Psidium* species. Furthermore, the high cross-transferability rate of synthesized g-SSR would reduce the additional expenses on development of species-specific microsatellite markers in genus *Psidium*.

## Genetic diversity and population structure studies

Several expeditions have been made to collect valuable natural variabilities distributed over the major guava growing regions in India. Furthermore, several exotic cultivars [79] and wild *Psidium* species have also been introduced to augment the germplasm in India [14, 80]. These guava germplasms have been conserved at the various national research centres in their field gene banks. Thereafter, these genotypes have been scientifically characterized, an essential step for their efficient management and utilization in the varietal improvement programmes. Guava is one of the perennial tree species and maintaining a single tee in field gene bank

involved land, labours and cost thus, duplicates or misnomers should be discarded to reduce the cost of germplasm management. The genetic characterization of the guava germplasm not only removes the duplicates and/ or misnomers but also helps in the selection of the diverse parent genotypes for using in the varietal improvement programmes. The various molecular markers have been used for the characterization of crop species of which simple sequence repeats (SSR) markers are one of the choicest marker systems due to its high polymorphism, co-dominant nature, highly reproducible, high rate of cross-transferability and easy to score *etc.* [27, 81]. In the past, Coser et al. [18]; Kherwar et al. [64]; Ma et al. [65]; Sitther et al. [76]; Viji et al. [82] and Kanupriya et al. [83] have efficiently characterized the guava germplasm using a set of SSR loci and deciphered the extent of genetic diversity. In the present investigation, 40 *Psidium* genotypes constituted by different cultivars/ cultigen, hybrids and wild *Psidium* species was undertaken for the genetic characterization using 26 developed genomic SSR loci. The N-J tree clustered the 40 guava genotypes into six clades with two outgroups which indicated high degree of their genetic diverseness. The developed genomic SSR loci clearly differentiated the cultivated and wild guava genotypes. Most of the wild *Psidium* species selected in this study were grouped in a common clade, which illustrated their genetic distinctness from other cultivated guava genotypes. Earlier, Sitther et al. [76] also pointed out the polymorphic genomic SSR markers developed by Risterucci et al. [30] could enable genetic differentiation of cultivars and wild guava genotypes. Different guava variants were selected to assess degree of relatedness as well as their genetic diverseness among parent and other guava genotypes. The N-J tree amply illustrated that Allahabad Safeda variant to be closely related to cv. Allahabad Safeda, while Hisar Surkha variant had its distinctness to the cv. Hisar Surkha. Further, the *Psidium chinensis* 1, which has pink pulp colour showed distinction from the *Psidium chinensis* 2 (white pulp colour) including the other wild *Psidium* species. The three open-pollinated seedlings of *P. chinensis* 2 were also assessed for their genetic relationship with the selected guava genotypes including the maternal parent. All the three open-pollinated seedlings did not group with maternal parent but grouped into three different clades of the N-J tree. The distinctness of open-pollinated seedlings might be due to the involvement of pollen source from different guava genotypes. The reproductive biology of guava frequently allows cross-pollination, which is around 41% [1, 3]. Thus, guava genotypes derived through seedling population have substantial variability [84], which allows selection of genotype(s) with desirable horticultural traits [85]. In the present study also the developed genomic SSR loci could efficiently deciphered the distinctness of genetically diverse guava genotypes/ species. However, genomic SSR loci did not group the guava genotypes according to their pulp colour. The moderate level of expected heterozygosity (0.53) was observed among the studied *Psidium* guava genotypes, while the observed heterozygosity was low (0.29). Thus, moderate deviation between the expected heterozygosity and observed heterozygosity. Previously, Sitther et al [76] illustrated that large differences between expected heterozygosity and observed heterozygosity among the studied guava accessions suggested the possibility of high inbreeding depression or founder effect during the domestication of crop species. The partial compatibility was observed among the results of cluster analysis, population structure and principal coordinate analysis.

The model-based population structure analysis is one of the choicest tools for the genetic categorization of any plant species including the perennial fruit tree species [86], which could be successfully utilized in conservation and optimization of the collected germplasm [87]. In the recent past, model-based population structure analysis was used for genetic differentiation of the guava germplasm. For instance, Kherwar et al. [64] genetically categorized the 36 guava varieties including wild species into five genetic groups using the population structure analysis in India. Earlier, Sitther et al. [76] genetically differentiated the guava germplasm collected at

USDA (U.S. Department of Agriculture) National Plants Germplasm System, Hilo, Hawaii using population structure analysis. Similarly, Model-based population structure analysis was performed in the presented investigation to genetically categorize the selected 40 guava genotypes including some *Psidium* species. However, the guava genotypes selected in this study did not group according to their pulp colour into different populations. Similar finding was also reported by Kherwar et al. [64]. The *Psidium* genotypes selected in the present study were grouped into three genetic groups. Furthermore, PCoA partly revalidated this genetic grouping of the guava genotypes including the wild species. The high allele-frequency divergence (>0.05) recorded between the different guava population groups indicated high genetic diversity amongst them. Similarly, AMOVA also revealed high genetic diversity among the populations, among the individuals and within the individual of the populations. Earlier, Kherwar et al. [64] had also reported a marginal level (6%) of AMOVA among the populations. The high genetic diversity among the populations in the present investigation may be due to the diverse germplasm representing distinct guava genotypes, namely, commercial cultivars, exotic cultivars, cultigen, open-pollinated seedling varieties and wild species rather than the single source of variability, *i.e.* open-pollinated varieties. Furthermore, the unique and rare alleles of SSR loci are providing the prospect for maintaining a high degree of genetic diversity [47, 88]. In the present study, 7 rare and 3 unique alleles were detected among the *Psidium* genotypes using 26 developed g-SSR loci, which additionally provided the prospect of maintaining the high genetic diversity amongst them. Thus, the existing high genetic diversity could be valuable genetic resource for the guava improvement programmes. These diverse genetic resources could serve as the parent source for broadening the genetic base of the existing cultivars. Furthermore, the developed 26 g-SSR loci were quite efficient and informative for genetic characterization and population structure analysis of the selected guava genotypes including wild species.

## Conclusion

Genomic library enrichment techniques could be successfully explored for the development of novel genomic SSR markers in guava (*Psidium guajava* L.). A total of 38 novel g-SSR loci were designed and synthesized; validated through the genetic diversity and population structure studies among the diverse guava genotypes including the wild guava species. The substantial genetic diversity parameters were recorded among the developed SSR markers. These markers could efficiently characterize and differentiate the population sub-groups of the guava genotypes including wild species. Thus, the developed g-SSRs were found efficient for the genetic diversity and population structure analyses of the selected guava genotypes. The developed SSRs showed high rate of cross-transferability among the wild guava species. Thus, we could add novel genomic microsatellite markers as an important genomic resource in guava, which would be efficiently utilized in conducting basic genetic studies on guava.

## Acknowledgments

The senior author duly acknowledges the Director and Joint Director (Research), ICAR-IARI, New Delhi. The authors also acknowledge the Director, ICAR-NBPGR, New Delhi for providing the laboratory facilities.

## Author Contributions

**Conceptualization:** Sanjay Kumar Singh, Rakesh Singh.

**Data curation:** Rakesh Singh.

**Formal analysis:** Chavlesh Kumar, Rakesh Singh.

**Funding acquisition:** Rakesh Singh.

**Investigation:** Rakesh Singh.

**Methodology:** Ritu Paliwal.

**Project administration:** Rakesh Singh.

**Resources:** Amit Kumar Goswami, A. Nagaraja.

**Software:** Rakesh Singh.

**Supervision:** Rakesh Singh.

**Validation:** Chavlesh Kumar, Ramesh Kumar, Ritu Paliwal.

**Visualization:** Rakesh Singh.

**Writing – original draft:** Chavlesh Kumar, Ramesh Kumar, Rakesh Singh.

**Writing – review & editing:** Sanjay Kumar Singh, Rakesh Singh.

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
