## [Decision Letter · Decision Letter 0]

12 Jun 2020

PONE-D-20-11182

Development of novel g-SSR markers in guava (Psidium guajava L.) cv. Allahabad Safeda and their application in genetic diversity, population structure and cross species transferability

PLOS ONE

Dear Dr. Singh,

Thank you for submitting your manuscript to PLOS ONE. After careful consideration, we feel that it has merit but does not fully meet PLOS ONE’s publication criteria as it currently stands. Therefore, we invite you to submit a revised version of the manuscript that addresses the points raised during the review process.

The two reviewers suggested many minor copy editing changes and improvement in the writing part of the manuscript. These changes are required to improve the quality of the manuscript. Therefore, authors should incorporate all the changes required in the manuscript and submit the revised version of the manuscript for further evaluation. The quality of representation of scientific data and Figures, Tables should be improved as per the required minimum standard of the journal.   

We look forward to receiving your revised manuscript.

Kind regards,

Swarup Kumar Parida, Ph.D.

Academic Editor

PLOS ONE

Journal Requirements:

4. Thank you for stating the following financial disclosure: "No"

Reviewers' comments:

Reviewer's Responses to Questions

**Comments to the Author**

1. Is the manuscript technically sound, and do the data support the conclusions?

Reviewer #1: Yes

Reviewer #2: Yes

2. Has the statistical analysis been performed appropriately and rigorously? 

Reviewer #1: Yes

Reviewer #2: Yes

3. Have the authors made all data underlying the findings in their manuscript fully available?

Reviewer #1: Yes

Reviewer #2: Yes

4. Is the manuscript presented in an intelligible fashion and written in standard English?

Reviewer #1: Yes

Reviewer #2: Yes

5. Review Comments to the Author

Reviewer #1: Manuscript has been written quite satisfactorily in light of the above literature. However, there are few typos in the manuscript, especially, regarding space before/ after comma. Further, 'Introduction', especially, first two paragraphs may be drastically shortened.

Reviewer #2: I would like to appreciate the authors (researchers) for their sincere efforts in adding useful genomic information to guava crop. They have used sound methodologies, software and statistical analysis for concluding their results.

Manuscript is well written. Most of the information provided by them is self explanatory.

One correction in abstract, line no.31 It is AMOVA, mistakenly written as AMOA

6. PLOS authors have the option to publish the peer review history of their article (what does this mean?). If published, this will include your full peer review and any attached files.

Reviewer #1: No

Reviewer #2: No

---

## [Author Response · Author response to Decision Letter 0]

3 Jul 2020

Response to Reviewers

Reviewer #1: Manuscript has been written quite satisfactorily in light of the above literature. However, there are few typos in the manuscript, especially, regarding space before/ after comma. Further, 'Introduction', especially, first two paragraphs may be drastically shortened.

Reply: Authors are highly thankful to reviewer to his encouraging comments and critical review of the manuscript. All the typos especially the space before and after comma has been corrected. The Introduction part has been modified and reduced as suggested by the reviewer.

Reviewer #2: I would like to appreciate the authors (researchers) for their sincere efforts in adding useful genomic information to guava crop. They have used sound methodologies, software and statistical analysis for concluding their results.

Manuscript is well written. Most of the information provided by them is self-explanatory.

Reply: Authors would like to thank reviewer for his kind words and encouragement.

One correction in abstract, line no.31 It is AMOVA, mistakenly written as AMOA

Reply: The correction suggested has been incorporated in the manuscript.

---

## [Decision Letter · Decision Letter 1]

29 Jul 2020

Development of novel g-SSR markers in guava (Psidium guajava L.) cv. Allahabad Safeda and their application in genetic diversity, population structure and cross species transferability

PONE-D-20-11182R1

Dear Dr. Singh,

We’re pleased to inform you that your manuscript has been judged scientifically suitable for publication and will be formally accepted for publication once it meets all outstanding technical requirements.

Kind regards,

Swarup Kumar Parida, Ph.D.

Academic Editor

PLOS ONE

Additional Editor Comments (optional):

Reviewers' comments:

Reviewer's Responses to Questions

**Comments to the Author**

1. If the authors have adequately addressed your comments raised in a previous round of review and you feel that this manuscript is now acceptable for publication, you may indicate that here to bypass the “Comments to the Author” section, enter your conflict of interest statement in the “Confidential to Editor” section, and submit your "Accept" recommendation.

Reviewer #1: All comments have been addressed

Reviewer #2: All comments have been addressed

2. Is the manuscript technically sound, and do the data support the conclusions?

Reviewer #1: Yes

Reviewer #2: Yes

3. Has the statistical analysis been performed appropriately and rigorously? 

Reviewer #1: Yes

Reviewer #2: Yes

4. Have the authors made all data underlying the findings in their manuscript fully available?

Reviewer #1: Yes

Reviewer #2: Yes

5. Is the manuscript presented in an intelligible fashion and written in standard English?

Reviewer #1: Yes

Reviewer #2: Yes

6. Review Comments to the Author

Reviewer #1: The suggested changes (as pointed out by both the reviewers) have been incorporated satisfactorily by the authors.

Reviewer #2: All the corrections were included by the authors. Now the manuscript is in a proper shape. It is a good piece of information for other researchers.

7. PLOS authors have the option to publish the peer review history of their article (what does this mean?). If published, this will include your full peer review and any attached files.

Reviewer #1: **Yes: **Hare Krishna

Reviewer #2: No

---

## [Editor Report · Acceptance letter]

3 Aug 2020

PONE-D-20-11182R1 

Development of novel g-SSR markers in guava (Psidium guajava L.) cv. Allahabad Safeda and their application in genetic diversity, population structure and cross species transferability 

Dear Dr. Singh:

I'm pleased to inform you that your manuscript has been deemed suitable for publication in PLOS ONE. Congratulations! Your manuscript is now with our production department. 

Kind regards, 

on behalf of

Dr. Swarup Kumar Parida 

Academic Editor

PLOS ONE